# Rheometer Evidences for the Co-Curing Effect of a Bismaleimide in Conjunction with the Accelerated Sulfur on Natural Rubber/Chloroprene Rubber Blends

**DOI:** 10.3390/polym13091510

**Published:** 2021-05-07

**Authors:** Marek Pöschl, Shibulal Gopi Sathi, Radek Stoček, Ondřej Kratina

**Affiliations:** Centre of Polymer Systems, Tomas Bata University in Zlín, Třida Tomáše Bati 5678, 760 01 Zlín, Czech Republic; poschl@utb.cz (M.P.); stocek@utb.cz (R.S.); okratina@utb.cz (O.K.)

**Keywords:** rubber, curing, bismaleimide, tensile strength, Diels–Alder reaction

## Abstract

The rheometer curing curves of neat natural rubber (NR) and neat chloroprene rubber (CR) with maleide F (MF) exhibit considerable crosslinking torque at 180 °C. This indicates that MF can crosslink both these rubbers via Alder-ene reactions. Based on this knowledge, MF has been introduced as a co-crosslinking agent for a 50/50 blend of NR and CR in conjunction with accelerated sulfur. The delta (Δ) torque obtained from the curing curves of a blend with the addition of 1 phr MF was around 62% higher than those without MF. As the content of MF increased to 3 phr, the Δ torque was further raised to 236%. Moreover, the mechanical properties, particularly the tensile strength of the blend with the addition of 1 phr MF in conjunction with the accelerated sulfur, was around 201% higher than the blend without MF. The overall tensile properties of the blends cured with MF were almost retained even after ageing the samples at 70 °C for 72 h. This significant improvement in the curing torque and the tensile properties of the blends indicates that MF can co-crosslink between NR and CR via the Diels–Alder reaction.

## 1. Introduction

It is well-known that natural rubber (NR) is a polymer of isoprene (2-methyl-1, 3-butadiene) and chloroprene rubber (CR) is a polymer of 2-chloro-1, 3-butadiene. Therefore, the main structural difference between NR and CR is that a methyl group in NR is substituted by a chlorine atom in CR. Because of the presence of this electronegative chlorine atom, CR has many unique properties such as improved heat, oil, ozone, and chemical resistance. Moreover, it has better resilience and weather resistance compared to NR [1,2]. It has been reported in the literature that commercial-grade polychloroprene comprises four isomeric forms such as trans-1, 4-polychloroprene (80–90%), cis-1, 4-polychloroprene (5–15%), 1, 2-polychloroprene (1–2%), and 3, 4-polychloroprene (3–4%) [3,4,5,6,7,8]. Out of these four isomeric forms, the 1, 2-isomer has been identified as the major isomer responsible for the curing process because of its ability to undergo the allylic rearrangement of the tertiary chlorine atom [9,10,11]. The rearrangement of the 1, 2-isomer can occur on the heating of the neat polychloroprene. However, the rearrangement occurs much faster in the presence of zinc oxide (ZnO) [12]. It is well known that ZnO or ethylene thiourea (ETU) are used as the main crosslinking agents for CR either separately or in combination [13,14]. Several mechanisms have been reported in the literature concerning the curing of CR with ZnO or ETU [13,14,15]. The mechanism proposed by Vukov based on the theory of Kuntz et al. using model compounds is considered as the most appropriate mechanism for the crosslinking of CR with ZnO [16,17]. Apart from ZnO or ETU, other chemicals such as, tribasic lead sulphate; thiophosphoryl disulfides; dimethyl L-cystine; and cetyltrimethylammonium maleate (CTMAM) have also been used as curing agents for CR [18,19,20,21]. Recently, Dziemidkiewicz et al. have used certain metal acetylacetonate as pro-ecological crosslinking agents for CR based on the Heck coupling reaction [22].

Unlike CR, the electropositive methyl group adjacent to the double-bonded carbon atom in NR enhances the activity of the unsaturated double bonds in the backbone of the polymer chains. As a result, NR has inferior weather, ozone, and oil resistance. Moreover, the high-temperature performance of NR is also limited because of its degradation above 70 °C. However, due to the active double bond, NR can be cured with accelerated sulfur systems. Based on the accelerator to sulfur ratio, accelerated sulfur curing systems are classified as conventional vulcanization (CV), efficient vulcanization (EV), and semi-efficient vulcanization (SEV). Generally, the CV system is characterized by a high dosage of sulfur (2–3.5 phr) and a low dosage of accelerators (0.4–1.2 phr) [1,2]. Even gum NR cured with a CV system can exhibit a tensile strength (TS) of around 20 to 25 MPa due to its unique strain-induced crystallization behaviour [23,24,25]. Blending between NR and CR can exploit certain unique properties of these individual rubbers. However, the structural disparity due to the electropositive methyl group in NR and the electronegative chlorine atom in CR makes the blending of NR and CR and the subsequent co-curing of the resultant blends very difficult. Therefore, a chemical that can co-cure both the NR and CR chains is essential for developing a compatible blend of NR/CR.

The rheometer is one of the key pieces of characterization equipment in the rubber industry and is used to check the feasibility of a new chemical as a curing agent in rubber compounds. Generally, the rheometer analysis gives a clear spectrum concerning the processing behaviors, such as the viscosity, scorch time, and optimum cure time of rubber compounds. Based on this knowledge, the compounder can select specific ingredients and determine the dosage of each ingredient needed to meet the required target. Moreover, with rheometer cure data, rubber scientists can quickly arrive at certain predictions concerning the crosslinking mechanisms based on the available theory. This may help the scientist establish the actual chemical reaction mechanisms involved in the curing process at the molecular level using advanced characterization techniques such as differential scanning calorimetry (DSC), nuclear magnetic resonance (HNMR), and infrared (IR) spectrometry, etc. From our previous experimental investigation, it has been observed that bismaleimide can react with halogenated rubbers during curing [26,27,28,29], and can also interact with NR/CV and butadiene rubber (BR)/CV systems via Diels–Alder and Alder-ene reactions [30,31,32]. Sadao Inoue filed a patent based on a chloroprene rubber and bismaleimide/ZnO composition for the development of vulcanizate with a high degree of crosslink density [33].

In the present investigation, we explored the curing behavior of a 50/50 blend of NR/CR with different contents of MF using a rheometer. The swelling behavior and mechanical properties of the vulcanizate derived from these blends were also evaluated. To the best of our knowledge, no reports are available in the literature concerning the curing behavior of an NR/CR blend with MF.

## 2. Materials

Natural rubber (standard Vietnamese rubber with a Mooney viscosity ML (1 + 4) at 100 °C: 60 ± 5) supplied by Binh Phuoc, Vietnam under the trade name SVR CV60 and chloroprene rubber (Neoprene 9243P, DuPont elastomer with a Mooney viscosity ML (1 + 4) at 100 °C: 87) were used as the base elastomers. Maleide F (MF) is a combination of 75% N, N’-meta phenylene dimaleimide and a 25% blending agent was procured from Krata Pigment, Tambov, Mentazhnikov, Russia. The chemical structure of MF is shown in Figure 1. Other ingredients such as sulfur; n-cyclohexyl-2-benzothiazole sulfenamide (CBS); stearic acid; zinc oxide (ZnO); and magnesium oxide (MgO) were purchased from Sigma-Aldrich, Czech Republic.

### 2.1. Preparation of Rubber Compounds

The formulation of the mixes with designations are displayed in Table 1. All the compounds were prepared using an internal mixer (Brabender Plastograph, GmbH & Co, KG, Duisburg, Germany) with a chamber volume of 50 cc. A fill factor of 0.8 was taken for the efficient mixing of the ingredients. To prepare the CR-based compound, the neat CR was masticated at 50 °C under 50 rpm for 2 min. To this, the ZnO, MgO, stearic acid, and MF were added, and the mixing was continued under the same rotor speed and temperature for another 2 min. After the mixing, the compound was discharged and homogenized using a two-roll mill. To prepare the blend-based compounds, the individual rubbers were masticated separately for 2 min under the same processing conditions. The pre-masticated rubbers were mixed for 1 min. To this, the ZnO, MgO, stearic acid, and MF were added, and the mixing was continued for 2 more minutes. Finally, the sulfur and CBS were added and mixed for an additional minute. After the mixing, the compound was discharged and homogenized using a two-roll mill. It was then molded into sheets with a thickness of 2 mm by applying a constant force of 200 N using a compression molding heat press LaBEcon 300 (Fontijne Presses, Delft, Netherlands) for the respective cure time obtained from the rheometer cure data at 180 °C.

### 2.2. Characterization

#### 2.2.1. Cure Characteristics

Maximum torque: *M_H_,* minimum torque: *M_L_,* the difference between maximum and minimum torque: Δ*M*, scorch time: *T_S_*_2_, optimum cure time: *T*_90_ (the time required for the torque to reach 90% of the maximum torque) of the rubber compounds were determined from the cure curves from a moving die rheometer (MDR-3000, MonTech, Buchen, Germany) at 180 °C as per ASTM D 5289. The cure rate index (CRI), a measure of the rate of curing, was calculated using Equation (1).
CRI = 100/(*T*_90__-*S*__2_).(1)

#### 2.2.2. Swelling Behavior

Samples with a diameter of 20 mm and a thickness of 2 mm with an initial weight (*W_i_*) were swelled in toluene at room temperature until they reached an equilibrium state of swelling. The swelled samples were then taken out and wiped off the adhered toluene from the surface using a filter paper, and the weights (*W_s_*) were immediately recorded. The swollen samples were dried at room temperature (20 °C) for 24 h, and ensured that the absorbed toluene is completely expelled out. Then, we measured the dried weight of the samples (*W_d_*). From the values of *W_i_*, *W_s_,* and *W_d_* the percentage swelling and the swell ratio were calculated using Equations (2) and (3), respectively [34,35].
(2)$$Swelling\;\left(\%\right)=\frac{W_{s}-W_{i}}{W_{S}}\times 100$$
(3)$$Swell\;ratio\left(Q\right)=\frac{W_{s}}{W_{d}}-1$$

#### 2.2.3. Mechanical (Tensile) Properties 

The stress–strain behavior and the corresponding tensile properties of the vulcanizates were measured using a universal testing machine (Testometric M350, Testometric Company, Ltd. Rochdale, UK). The testing was performed under ambient conditions at a crosshead speed of 500 mm/min as per ISO 37 using S2 type specimen with a thickness of 2 mm. The results were reported at an average of six tested specimens. The properties of the cured samples were also measured after ageing at 70 °C and 100 °C for 72 h using a forced air circulating oven.

#### 2.2.4. Hardness Testing

Cured samples having smooth surfaces were used to measure the indentation hardness using a Shore-A hardness tester (Bareiss Durometer, Oberdischingen, Germany) as per ASTM D 2240. Indentations were made on different areas of the samples by applying constant pressure for 15 s. Six readings were taken from different areas of the sample and we reported the average value.

## 3. Results and Discussion 

### 3.1. Curing Behavior of Neat CR with ZnO and MF

Represented in Figure 2 are the curing curves of neat CR with ZnO, MF, and a combination of ZnO/MF at 180 °C for 1 h. Their cure characteristics are depicted in Table 2. The cure curve of CR/ZnO (M-1) exhibits a fast curing reaction, as evident from the short *ts_2_* value (0.65 min). However, after a rapid initial curing reaction, the cure curve turned into a marching modulus behavior and ended up with a maximum torque of 5.80 dNm at the given curing time. As a result, the *t*_90_ (40.2 min) value was higher than expected. Several mechanisms have been proposed in the literature to explain the curing behavior of CR with ZnO. Out those, a cationic mechanism proposed by Vukov is widely accepted [21]. As per this mechanism, the 1, 2-isomer of CR undergoes a rearrangement (isomerization) upon heating above 160 °C. The rearranged 1, 2-isomer then produces a conjugated diene in the presence of ZnO. The rearranged 1, 2-isomer and the in situ-formed diene catalyzed by ZnCl_2_ produce the crosslinks.

The cure curve of M-1 shows a relatively low Δ*M* (4.42 dNm) value, indicating that the extent of curing of CR with ZnO is not high enough. On the other hand, the curing of CR with MF alone (M-2) progressed at a slow pace and exhibited a marching modulus curing right from the beginning until the end of the given curing time. As a result, the *ts*_1_ (2.43 min) and *ts*_2_ (4.42 min) values were higher compared to M-1. Though the curing curves of M-1 and M-2 exhibited a marching modulus curing behavior, the extent of vulcanization was greatly increased in M-2 as the cure time progressed. For instance, the Δ*M* (extent of vulcanization) after 30 min of curing in M-1 was 3.70 dNm. At the same time, the ΔM in M-2 after 30 min of curing was around 114% higher than M-1. This dramatic improvement in Δ*M* indicates that MF alone can substantially crosslink the CR chains. Since there are no other ingredients in the system other than CR and MF, one the plausible crosslinking reactions might be the Alder-ene reaction between the rearranged 1,2-isomer of CR and MF, as depicted in Figure 3. 

It was interesting to note that the curing of CR with a combination of ZnO and MF (M-3) exhibits an initial rapid reaction (*ts*_2_: 0.60 min) with a plateau-type curing behavior. As a result, the *t*_90_ value of M-3 (20.63 min) was much lower compared to M-1 (*t*_90_: 40.23 min) and M-2 (*t*_90_: 41.58 min). Moreover, the extent of curing was also higher compared to M-1 and M-2. For instance, the Δ*M* generated after 10 min of curing in M-3 was around 152% higher than M-1 and 70% higher than M-2 under the same curing conditions. It has been reported that the 1, 2-isomer of CR undergoes a rearrangement and subsequently produces a conjugated diene in the presence of ZnO [12]. Therefore, it is reasonable to believe that one of the reasons behind the synergistic curing behavior in M-3 might be the Diels–Alder reaction between the in situ-formed diene and the maleimide moieties of MF, as shown in Figure 4a.

### 3.2. Curing Behavior of NR/CR Blend in the Presence of MF in Conjunction with Accelerated Sulfur

Represented in Figure 5 are the cure curves of NR/CR blends corresponding to the mixes 4–7 at 180 °C for 1 hr. The cure curve of the blend without MF (M-4) exhibits a scorch time of 3.86 min with a Δ*M* value of 2.39 dNm. The addition of 1 phr MF reduces the scorch time of M-4 from 3.86 min to 2.54 min and improves the Δ*M* value by 62%. Similarly, the addition of 3 phr MF improves the Δ*M* value of M-4 by 236%, which further rose to 289% with the addition of 5 phr MF. Here, it is interesting to note that although there are not many differences in the scorch time between M-4 and the rest of the mixes, the addition of MF gradually increases the optimum cure time as the content of MF increases. For instance, the *t*_90_ of M-4 was 4.62 min, which became 12.44 min after the addition of 3 phr MF and 18.36 min with the addition of 5 phr MF. This might be due to the slightly marching modulus curing behavior of the mixes containing MF. From this study, it has been observed that CR undergoes a synergistic curing behavior in the presence of a combination of ZnO and MF. One of the reasons for this was suspected to be the Diels–Alder reaction between the in situ-formed diene from the isomerized CR and the malemide moieties of MF as shown in Figure 4a. From our previous experimental investigation, it has been identified that MF can act as an anti-reversion agent during the curing of NR and BR with a CV system [30,31,32]. One of the plausible mechanisms proposed to explain the anti-reversion ability of MF was also the Diels–Alder reaction between the in situ formed diene from the NR/CV system and the maleimide moieties of MF. Based on these experimental inferences, it is reasonable to believe that the three types of reactions given in Figure 4a–c might be possible during the curing of mixes 4–7. The occurrence of any of these three reactions can improve the overall crosslink density of the blend system. However, the co-curing reaction given in Figure 4c is essential for enhancing the compatibility between NR and CR.

### 3.3. Swelling Behavior

It is well known that the crosslinked rubbers with a tight network structure generally show high swelling resistance. The percentage swelling and the swell ratio as per Equations (2) and (3) were calculated for the blends (M-4 to M-7), and the results are given in Table 3.

For comparative purposes, the swell ratio of M-1, M-2, and M-3 were also given. The blend with no MF (M-4) exhibits a solvent uptake of around 551%, corresponding to a swell ratio of 6.5. As the content of MF increased, the percentage swelling gradually decreased. This means that a higher concentration of MF produces more crosslinked points between the polymer chains as per the reactions proposed in Figure 4, thereby enhancing the extent of crosslinking and the network density. The Δ*M* values obtained from the rheometer cure data were also in line with this swelling behavior. However, the rheometer data and the swelling behavior do not give a clear indication of which reaction represented in Figure 4 is predominant during the curing of blends containing MF.

### 3.4. Mechanical Properties of the Blends

To understand the crosslinking effects of MF on the mechanical properties, the tensile strength (TS), the elongation at break (EB), the modulus at a different percentages of elongation, and the hardness of the cured blends were evaluated with different contents of MF, and the results are represented in Table 4.

For a comparative evaluation, the tensile properties of the vulcanizates of M-1 and M-3 are also given in Table 3. The vulcanizate of M-1 gives a TS of 5.32 MPa with an EB of 292%. Both the TS and the EB were reduced to 2.84 MPa and 154%, respectively, when CR was cured with a combination of ZnO and MF (M-3). The cured network of M-1 might not be strong enough because the network mainly composed of carbon–carbon crosslinks as per the reaction mechanism proposed by Vukov. Therefore, the cured network of M-1 might have certain flexibility to transfer the tensile load, and thereby exhibit a relatively high TS and EB compared to the vulcunizate of M-3. The additional crosslinks formed in the vulcanizate of M-3 owing to the proposed Diels–Alder reaction shown in Figure 4a, the cured network of M-3 will be rigid and strong. As a result, the transfer of tensile load becomes difficult and hence shows a low TS and strain at break. However, the strong network structure in M-3 significantly enhances its hardness and modulus. It was interesting to note that the vulcanizate of the blend with no MF (M-4) gives a TS of 3.07 MPa and an EB of 490%. However, after the incorporation of 1 phr MF, the TS of the blend vulcanizate (M-5) suddenly improved and was three times higher than M-4. Both the modulus and hardness of M-5 were also significantly improved with the addition of even 1 phr MF. The improved mechanical properties of the blends cured with MF in conjunction with the accelerated sulfur give a strong indication concerning the co-crosslinking reaction between NR and CR, as proposed in Figure 4c. It is worth noting that the TS of the blends did not improve further beyond 1 phr MF. That being said, the modulus and hardness were significantly improved up to the addition of 5 phr MF. To understand the strength of the blend vulcanizate, the abovementioned tensile properties were also evaluated after ageing them at 70 °C and 100 °C for 72 h. The results are depicted in Figure 6a–c.

From the results, it is clear that the properties of the unaged and aged samples at 70 °C for 72 h were comparable. However, the properties, particularly the TS were almost 3 times lower when the blends were aged at 100 °C for 72 h. This might be due to the degradation of the NR phase in the blend during the ageing process at 100 °C.

## 4. Conclusions

The rheometer investigation of the curing of neat CR with a combination of ZnO and MF at 180 °C exhibits a synergistic curing curve with a plateau type cure behavior. In light of the mechanism proposed by Vukov, herein we propose that the Diels–Alder reaction between the in situ-formed diene from CR/ZnO and the maleimide moieties of MF were responsible for this synergistic curing behavior. Similarly, a synergistic curing behavior was also observed in the rheometer study when a 50/50 blend of NR/CR was cured with MF in conjunction with a conventional type accelerated sulfur (CV) system. The tensile properties and the hardness of the blends were significantly improved after curing with MF. A co-vulcanization reaction between the dienes generated from the CR and the NR phases of the blends with either ends of the maleimide moieties via Diels–Alder reaction was proposed to substantiate the observed synergistic curing behavior and the improved tensile properties of the blends. The rheometer evidence concerning the curing of CR with ZnO/MF demands an advanced investigation for further confirming the proposed Diels–Alder reaction.

## Figures and Tables

**Figure 1 polymers-13-01510-f001:**
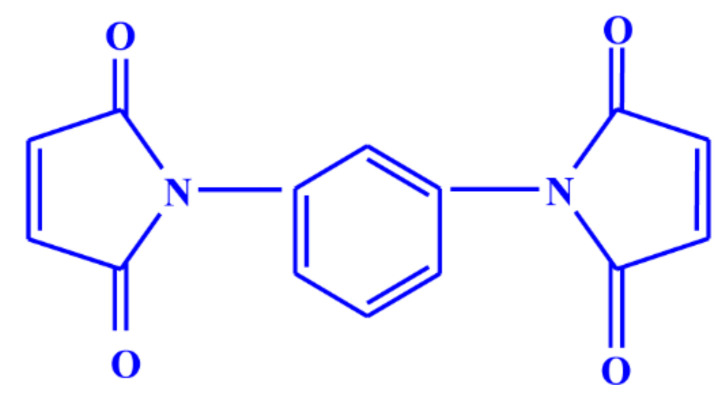
Chemical structure of N, N’-meta phenylene dimaleimide (Maleide F).

**Figure 2 polymers-13-01510-f002:**
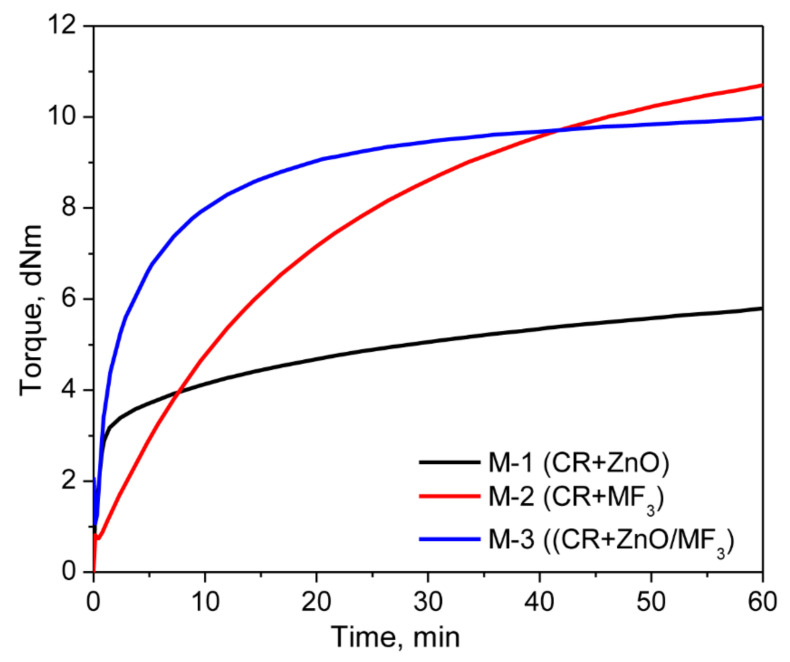
Cure curves of the mixes M-1, M-2, and M-3.

**Figure 3 polymers-13-01510-f003:**
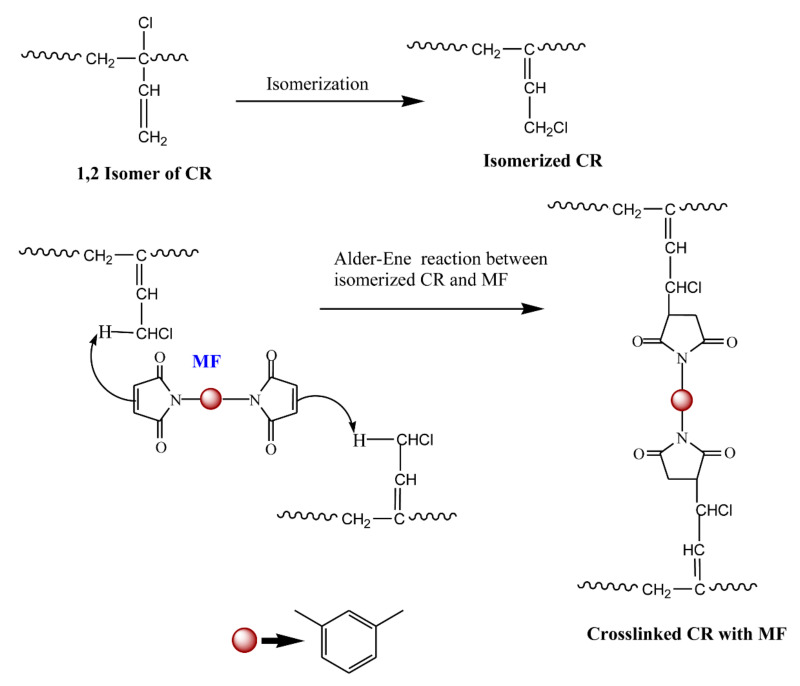
Plausible reaction mechanism for the curing behavior of CR with MF.

**Figure 4 polymers-13-01510-f004:**
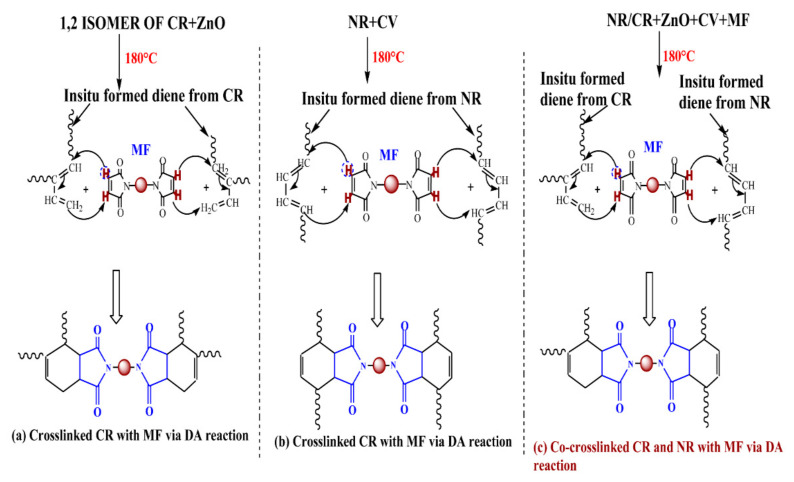
Plausible reactions in NR/CR blend during curing with MF in conjunction with the accelerated sulfur system.

**Figure 5 polymers-13-01510-f005:**
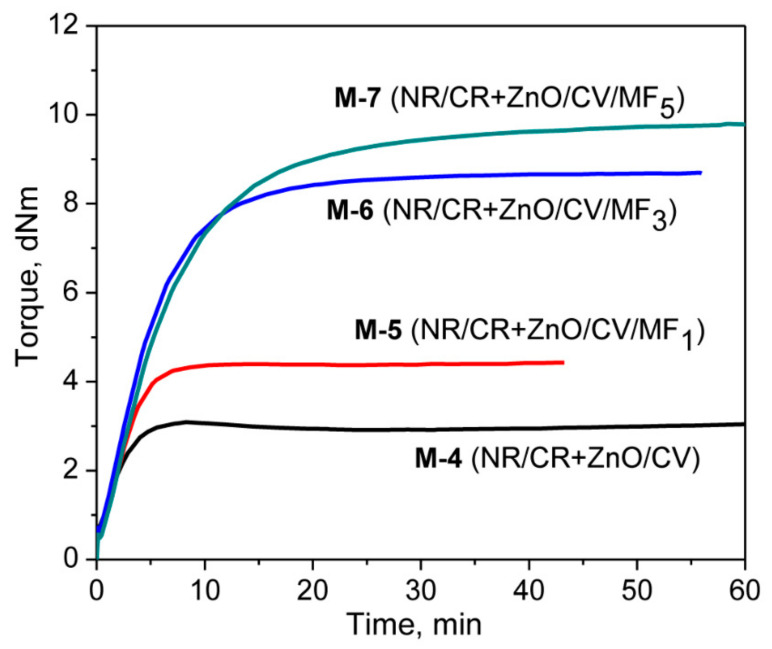
Cure curves of the mixes M-4, M-5, M-6, and M-7.

**Figure 6 polymers-13-01510-f006:**
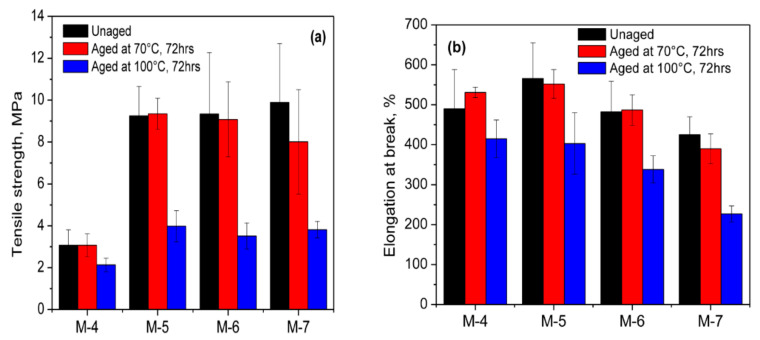
Tensile properties of blend vulcanizates (**a**) tensile strength, (**b**) elongation at break, and (**c**) modulus at 100% elongation before and after ageing.

**Table 1 polymers-13-01510-t001:** Formulation of the mixes.

Ingredients	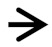	NR	CR	ZnO	MgO	Stearic Acid	Sulfur	CBS	Maleide F
Mix Code	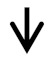
M-1		-	100	5	4	0.5	-	-	-
M-2		-	100	-	-	-	-	-	3
M-3		-	100	5	4	0.5	-	-	3
M-4		50	50	5	2	1.25	1.25	0.25	-
M-5		50	50	5	2	1.25	1.25	0.25	1
M-6		50	50	5	2	1.25	1.25	0.25	3
M-7		50	50	5	2	1.25	1.25	0.25	5

**Table 2 polymers-13-01510-t002:** Rheometer cure data of the mixes at 180 °C, 1 h.

Mix Code	*M_L_*(dNm)	*M_H_* (dNm)	Δ*M* (dNm)	*T_S_*_1_ (min)	*T_S_*_2_ (min)	*T*_90_ (min)	Cure Rate Index (min^−1^)
M-1	1.38	5.80	4.42	0.65	0.65	40.23	2.52
M-2	0.74	10.69	9.95	2.43	4.52	41.58	2.69
M-3	1.07	9.97	8.90	0.53	0.70	20.63	4.99
M-4	0.70	3.09	2.39	1.65	3.86	4.62	131.57
M-5	0.55	4.43	3.88	1.55	2.54	5.55	33.22
M-6	0.66	8.70	8.04	1.34	2.30	12.44	9.86
M-7	0.49	9.80	9.31	1.51	2.38	18.36	6.25

**Table 3 polymers-13-01510-t003:** Percentage swelling and swell ratio.

Mix Code	Swelling (%)	Swell Ratio (Q)
M-1	281.7 ± 0.1	2.94 ± 0.1
M-2	237.2 ± 0.02	2.35 ± 0.003
M-3	177.0 ± 0.07	1.78 ± 0.08
M-4	562.9 ± 0.08	6.55 ± 0.13
M-5	428.2 ± 0.03	4.57 ± 0.12
M-6	314.9 ± 0.13	3.29 ± 0.1
M-7	267.9 ± 0.06	2.79 ± 0.06

**Table 4 polymers-13-01510-t004:** Tensile properties and hardness of the mixes.

Mie	Tensile Strength (MPa)	Elongation at Break (%)	Modulus at 50% (MPa)	Modulus at 100% (MPa)	Modulus at 300% (MPa)	Hardness (Shore A)
M-1	5.32 ± 0.98	292 ± 63	0.91 ± 0.13	1.56 ± 0.26	4.03 ± 1.87	39
M-3	2.84 ± 0.20	155 ± 9.0	1.08 ± 0.08	1.83 ± 0.19	-	52
M-4	3.07 ± 0.74	490 ± 98	0.37 ± 0.03	0.54 ± 0.05	1.38 ± 0.26	24
M-5	9.25 ± 1.40	566 ± 89	0.62 ± 0.13	0.90 ± 0.10	2.55 ± 0.67	34
M-6	9.34 ± 2.93	482 ± 77	0.70 ± 0.11	1.07 ± 0.12	3.08 ± 0.48	42
M-7	9.89 ± 2.81	425 ± 45	0.85 ± 0.11	1.33 ± 0.18	4.32 ± 1.34	47

## Data Availability

The data presented in this study are available on request from the corresponding author.

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
