# Peer review of "Rheometer Evidences for the Co-Curing Effect of a Bismaleimide in Conjunction with the Accelerated Sulfur on Natural Rubber/Chloroprene Rubber Blends"

_polymers, 2021, doi:10.3390/polym13091510_

Round 1

Reviewer 1 Report

This is a fundamental research about the co-curing effect of a bismaleimide in conjunction with the accelerated-sulphur on natural rubber/chloroprene rubber blends. The authors have been focusing on using rheometer to study the curing/cross-linking process of natural rubber/chloroprene rubber blends, which does provide new insight into the field. However, I still have some minor concerns/suggestions before I can recommend it for publication.

(1) The error analysis of this work is not properly handled at some places. Some data presented have too many significant figures in standard deviation. For example, in table 3, "281.7±0.090" is not a reasonable expression. It should be "281.7±0.1". Similar rule applies to other data.

(2) While the main insight of this work focuses on using Bismaleimide as crosslinking agent, it would be good to include some discussions/comparison with other cross-linking agents for natural rubber.

(3) In the introduction sections, the authors directly go to natural rubber and chloroprene rubber, which may not be appealing to broad interest of perspective readership. I would like to suggest the authors start with an overview of engineering of natural product for improved property and applications, such as cellulose(DOI: 10.21967/jbb.v4i1.186; DOI: 10.12162/jbb.v4i4.014) and chitosan (DOI: 10.21967/jbb.v4i1.189; Journal of Forestry Engineering, 2020, 5(2): 76-81).

Based on the above concerns, I would like to suggest a minor revision.

Author Response

This is a fundamental research about the co-curing effect of a bismaleimide in conjunction with the accelerated-sulphur on natural rubber/chloroprene rubber blends. The authors have been focusing on using rheometer to study the curing/cross-linking process of natural rubber/chloroprene rubber blends, which does provide new insight into the field. However, I still have some minor concerns/suggestions before I can recommend it for publication.

Comment 1:  The error analysis of this work is not properly handled at some places. Some data presented have too many significant figures in standard deviation. For example, in table 3, "281.7±0.090" is not a reasonable expression. It should be "281.7±0.1". Similar rule applies to other data.

Response 1:  Thank you for indicating the mistakes. It has been corrected now.

Comment 2:  While the main insight of this work focuses on using Bismaleimide as a crosslinking agent, it would be good to include some discussions/comparison with other cross-linking agents for natural rubber.

Response 2: As suggested by the reviewer, some additions have been included in page 2. However, in this work, we have made a preliminary investigation to understand whether bismaleimide is useful as a potential chemical to co-crosslink between NR and CR in their blends. For a preliminary understanding, a rheometer investigation was conducted on NR/CR blend with bismaleimide alone and also in conjunction with ZnO/MgO (curing agent for CR) and sulfur/CBS (curing agent for NR).

Comment 3:  In the introduction sections, the authors directly go to natural rubber and chloroprene rubber, which may not be appealing to broad interest of perspective readership. I would like to suggest the authors start with an overview of engineering of natural product for improved property and applications, such as cellulose (DOI: 10.21967/jbb.v4i1.186; DOI: 10.12162/jbb.v4i4.014) and chitosan (DOI: 10.21967/jbb.v4i1.189; Journal of Forestry Engineering, 2020, 5(2): 76-81). Based on the above concerns, I would like to suggest a minor revision.

Response 3:  The introduction part is now slightly modified

Reviewer 2 Report

In this report the authors investigated the curing with ZnO and MF agent in natural rubber/chloroprene rubber blends mostly with a sealed blender. Also the mechanism of the increase in torque with MF agent in the composition can be explained by the Diels-Alder reaction mechanism.

This report is well formatted and organized though some improvements have to be made before it can be published. 

  • Language has to be improved:
  • Literature says that commercial…
  • Line83 kindkly

  • Line 194. M-2 is curing with ZnO along or with MF along? The text and legend on Figure 2 reads differently. Please verify. In the following text it reads like the legend is messed up in Figure 2
  • The purpose of this study should be elaborated more deliberately. Why the CR/NR blend is needed? And why we do not use MF agent if it provide faster curing during cross linking? It may not that obvious to the general audience.
  • This feels more like a detailed technical report other than a scientific report. The authors proposed the Diels-Alder reaction but only minimal explanation is provided. It feels like we only got this one option.  Is there any reaction control function that we can fit with experimental data as a proof? Or SEM/TEM images on morphology intermediately to show the reaction is actually taking place?

Author Response

In this report the authors investigated the curing with ZnO and MF agent in natural rubber/chloroprene rubber blends mostly with a sealed blender. Also the mechanism of the increase in torque with MF agent in the composition can be explained by the Diels-Alder reaction mechanism.

Comment 1: This report is well formatted and organized though some improvements have to be made before it can be published. 

  • Language has to be improved:
  • Literature says that commercial…
  • Line83 kindkly

Response: Thanks for indicating the corrections. It has been improved now.

Comment 2 : Line 194. M-2 is curing with ZnO along or with MF along? The text and legend on Figure 2 reads differently. Please verify. In the following text it reads like the legend is messed up in Figure 2

Response: We acknowledge our sincere thanks to the reviewer for indicating the mistake.  The legends in the figure have been corrected now.

Comment 3: The purpose of this study should be elaborated more deliberately. Why the CR/NR blend is needed? And why we do not use MF agent if it provide faster curing during cross linking? It may not that obvious to the general audience.

Response: From our previous rheometer investigations, it has been identified that MF alone and also in combination with accelerated-sulphur can vulcanize NR and there by significantly improves the physico-mechanical properties of the resultant vulcanizate [J. Polym. Res. 2016, 23, 237-248, EXPRESS Polym. Lett. 2020, 14, 838-847]. Moreover, it has also been noticed that bismaleimide can vulcanize halogenated butyl rubbers in the presence of ZnO [J. Polym. Res. 2018, 25, 108-121, Polym. Adv. Technol. 2017, 28, 742–753, Rubber Chem. Technol. 2019, 92, 110-128]. These experimental observations motivated us to explore the use of a bismaleimide (MF) as a co-vulcanizing agent for NR/CR blend. Use of MF as sole crosslinking agent has many limitations like marching modulus curing behaviour which has been discussed in the script.

Comment 4: This feels more like a detailed technical report other than a scientific report. The authors proposed the Diels-Alder reaction but only minimal explanation is provided. It feels like we only got this one option.  Is there any reaction control function that we can fit with experimental data as a proof? Or SEM/TEM images on morphology intermediately to show the reaction is actually taking place?

Response: We do agree with the reviewer's comments. This is the report of a preliminary scientific investigation concerning the possibility of MF as a co-crosslinking agent for the NR/CR blend. At present, we have only the rheometer data to support the fact that the proposed curing mechanism follows the Diels-Alder reaction. However, we are still working on this to elaborate on the proposed reaction mechanisms with more promising experimental evidence. Hopefully, we will be able to publish our findings in the next communication.

Reviewer 3 Report

The curing of neat CR with a combination of ZnO and 369 MF has been investigated in this manuscript. The Diels- Alder reaction between the in-situ formed diene from CR/ZnO and the maleimide moieties of MF was considered responsible for the synergistic curing behaviour. This work will provide guide for further investigation of the proposed Diels-Alder reaction. To present a high-quality publication, follow revisions are advised: 

  1. The language of English should be improved. There are several spelling and grammar mistakes.
  2. The curing of neat chloroprene rubber has been investigated and reviewed in some recent work. Please refer to Energy Storage Mater. 34 (2021) 107-127, Energy Storage Mater. 27 (2020) 279-296 and discuss in the introduction part.

Author Response

The curing of neat CR with a combination of ZnO and 369 MF has been investigated in this manuscript. The Diels- Alder reaction between the in-situ formed diene from CR/ZnO and the maleimide moieties of MF was considered responsible for the synergistic curing behaviour. This work will provide guide for further investigation of the proposed Diels-Alder reaction. To present a high-quality publication, follow revisions are advised: 

Comment 1: The language of English should be improved. There are several spelling and grammar mistakes.

Response: Proper modifications have been made accordingly

Comment 2: The curing of neat chloroprene rubber has been investigated and reviewed in some recent work. Please refer to Energy Storage Mater. 34 (2021) 107-127, Energy Storage Mater. 27 (2020) 279-296 and discuss in the introduction part.

Response: Thanks for the suggestion. We have gone through the suggested article by the reviewer. However, these articles are not related to the topic of this paper and therefore not included in the script as references.

Reviewer 4 Report

This is interesting research. However, the manuscript needs some modifications as follows:

  • The English language should be improved throughout the manuscript. Please write the manuscript in reporting style or using passive sentences (not active sentences, e.g. we, I, etc), see Abstract, Line # 14 " we introduced". This comment applies to the whole manuscript. Please revise.
  • The horizontal spaces in Table 4 should be removed.
  • Abstract and conclusions should be extended to include the results of mechanical properties especially the effect of ageing.

Author Response

This is interesting research. However, the manuscript needs some modifications as follows:

Comment 1: The English language should be improved throughout the manuscript. Please write the manuscript in reporting style or using passive sentences (not active sentences, e.g. we, I, etc), see Abstract, Line # 14 " we introduced". This comment applies to the whole manuscript. Please revise.

Response: Corrected as per the suggestion.

Comment 2: The horizontal spaces in Table 4 should be removed.

Response: It has been corrected now

Comment 3: Abstract and conclusions should be extended to include the results of mechanical properties especially the effect of ageing.

Response: It has been modified now

Reviewer 5 Report

This is a detailed study into the crosslinking of polychloroprene rubber utilising bismaleimide as the main agent coupled with synergy involving interactions with metal oxides. The methodologies are wide ranging with a good complete in depth evaluation of the chemical and physical processes involved. Its not a new field, far from it, such that the basis of the study is not new. However, the data and synergistic reactions provides some valuable outcomes. I might suggest the authors reference the PATENT literature as this is abundant with polychloroprene crosslinking via the same components i.e. 

JPS59109541A

Author Response

Comment : This is a detailed study into the crosslinking of polychloroprene rubber utilising bismaleimide as the main agent coupled with synergy involving interactions with metal oxides. The methodologies are wide ranging with a good complete in depth evaluation of the chemical and physical processes involved. Its not a new field, far from it, such that the basis of the study is not new. However, the data and synergistic reactions provides some valuable outcomes. I might suggest the authors reference the PATENT literature as this is abundant with polychloroprene crosslinking via the same components i.e. JPS59109541A

Response: Thank you for evaluating our work. The patent reference suggested by the reviewer has been included in the revised script